# Nonpharmacological Treatment Strategies for the Management of Canine Chronic Inflammatory Enteropathy—A Narrative Review

**DOI:** 10.3390/vetsci9020037

**Published:** 2022-01-20

**Authors:** Marco Isidori, Ronald Jan Corbee, Massimo Trabalza-Marinucci

**Affiliations:** 1Department of Veterinary Medicine, University of Perugia, Via San Costanzo 4, 06126 Perugia, Italy; massimo.trabalzamarinucci@unipg.it; 2Department of Clinical Sciences of Companion Animals, Faculty of Veterinary Medicine, Yalelaan 108, 3584 CM Utrecht, The Netherlands; r.j.corbee@uu.nl

**Keywords:** canine chronic inflammatory enteropathy, clinical nutrition, postbiotic, prebiotic, phytochemical, probiotic, synbiotic, faecal microbiota transplantation, stem cell therapy

## Abstract

Chronic inflammatory enteropathy (CIE) refers to a heterogeneous group of idiopathic diseases of the dog characterised by persistent gastrointestinal (GI) clinical signs. If conventional dietary treatment alone would be unsuccessful, management of CIE is traditionally attained by the use of pharmaceuticals, such as antibiotics and immunosuppressive drugs. While being rather effective, however, these drugs are endowed with side effects, which may impact negatively on the animal’s quality of life. Therefore, novel, safe and effective therapies for CIE are highly sought after. As gut microbiota imbalances are often associated with GI disorders, a compelling rationale exists for the use of nonpharmacological methods of microbial manipulation in CIE, such as faecal microbiota transplantation and administration of pre-, pro-, syn- and postbiotics. In addition to providing direct health benefits to the host via a gentle modulation of the intestinal microbiota composition and function, these treatments may also possess immunomodulatory and epithelial barrier-enhancing actions. Likewise, intestinal barrier integrity, along with mucosal inflammation, are deemed to be two chief therapeutic targets of mesenchymal stem cells and selected vegetable-derived bioactive compounds. Although pioneering studies have revealed encouraging findings regarding the use of novel treatment agents in CIE, a larger body of research is needed to address fully their mode of action, efficacy and safety.

## 1. Introduction

Chronic inflammatory enteropathy (CIE) is an umbrella term coined to define a group of diseases affecting the enteral system of the dog, whose diagnosis requires the exclusion of known digestive and extradigestive causes of chronic GI signs [1]. CIE is encountered in referral practice with a prevalence as high as 2% [2,3], and it is clinically classified upon treatment response to different therapeutic trials. Accordingly, four disease phenotypes have been identified, namely food-responsive (FRE), antibiotic-responsive (ARE), immunosuppressive-responsive (IRE) and nonresponsive (NRE) enteropathy [1]. It is generally accepted that term “idiopathic inflammatory bowel disease” (IBD) may be used in lieu of IRE, being the form of CIE that best resembles human IBDs, namely Crohn’s disease (CD) and ulcerative colitis (UC) [4].

Aside from FRE, for which dietary management alone has been proven effective to enable histological mucosal healing [5] and guarantees an adequate, long-lasting control of outward clinical signs [6], all other types of CIE classically mandate pharmacological treatment. Specifically, the therapeutic armamentarium used in the management of these ailments encompasses antibiotics (e.g., tylosin, metronidazole, oxytetracycline, rifaximin) and/or immunosuppressive drugs such as corticosteroids, cyclosporin A, azathioprine and chlorambucil [7].

Although characterised by a fairly high rate of clinical remission [8,9], reiterated or long-term administration of the aforementioned medications should be carefully weighed in light of their benefits and side effects. In this regard, numerous reports have disclosed the negative impact of antibiotics towards the gut microbiota, leading to significant drops in microbial diversity, evenness and species richness, i.e., dysbiosis [10,11,12]. Besides, there is strong evidence that the use of broad-spectrum antimicrobials is able to prompt the development of multidrug-resistant pathobionts in the dog, which poses a severe threat to both animal and human health [13,14]. Likewise, immunosuppressors are known to possess a plethora of side effects that may worsen the clinical burden of the disease [7]. Moreover, it must be stressed how some cases of CIE become less responsive to immunosuppressive treatment over time or might develop into intestinal neoplasms due to a sustained impairment of host defences [15,16].

The difficulty of combining the chronicity of the disease with the undesirable effects of currently available medications demands the development of new treatment protocols endowed with a good effectiveness and safety profile to improve the management of CIE. In this perspective, nonpharmacological interventions such as bacteriotherapy, cell therapy and the administration of nutraceuticals have garnered increasing attention over the last decade for their potential applications in companion animal gastroenterology. Remarkably, nonpharmacological agents may exert similar (e.g., anti-inflammatory, immunomodulatory or antimicrobial) biological effects compared to conventional treatments without competing for the same molecular targets [17]. The aforementioned premise is of utmost importance in enabling the development of additive and/or synergistic combination therapies that could guarantee a satisfactory control of the disease while using a lower dosage of prescribed drugs, thereby reducing their undesirable side effects. However, the use of a specific intervention should be endorsed by clinicians only when substantiated by scientific evidence. As such, the purpose of the current review is to critically appraise the state of the art regarding the main complementary and alternative therapies that show promise in the treatment of CIE, with special emphasis on those ones that manipulate the host’s intestinal microbiota, as well as providing an outlook for the near future.

## 2. Etiopathogenesis of CIE

From an academic standpoint, much effort has been made to gain a better understanding of CIE, and different etiological hypotheses have been brought forward. For instance, adverse food reactions (AFRs) have been identified as a potential causative agent for FRE, whereas ARE was traditionally compared to the small intestinal bacterial overgrowth of humans [16].

Irrespective of their being distinct maladies or different clinical exacerbations of the same disorder, the different forms of CIE are thought to be complex and multifactorial diseases [18,19,20]. Borrowing the universally accepted paradigm for CD and UC, the two variants of human IBD, CIEs may be the result of a deranged mucosal immune response towards microbiological and environmental antigens [21]. Key contributors to the immune dysregulation seen in dogs with CIE can be classified into a primary disturbance of host immunity or disruption of the intestinal epithelial barrier, as well as alterations to the intestinal autologous microflora composition [22].

### 2.1. Immune System

Several lines of evidence speak to the existence of innate immunity aberrations in dogs with CIE [23]. Pattern recognition receptors (PRRs) are a set of proteins, expressed differentially by most innate immune effector cells (e.g., dendritic cells, macrophages, neutrophils) but also intestinal epithelial cells, which recognise conserved molecular motifs common to various microorganisms (the so-called pathogen-associated molecular patterns) and are deemed to be key for the maintenance of host–microbial interaction within the gut mucosa [24].

The two major PRR families are toll-like receptors (TLRs) and nucleotide-binding oligomerisation domains (NODs) [25]. In a canine study, the mRNA expression of bacteria-responsive TLR-2, -4 and -9 was found to be increased in duodenal and colonic biopsies obtained from patients diagnosed with idiopathic IBD relative to healthy controls [26]. Another study reported a mild correlation between mRNA expression of TLR-2 and clinical disease activity in duodenal mucosal samples obtained from idiopathic IBD dogs [27]; however, TLR-2 expression was not correlated with histological severity of the disease. In a study investigating polymorphisms in canine TLR-2, -4 and -5 genes, it was concluded that three TLR-5 and two TLR-4 nonsynonymous single-nucleotide polymorphisms are likely to play a mechanistic role in idiopathic IBD pathogenesis in German shepherd dogs (GSDs) [28]. Later, the same authors demonstrated, in both in vitro and ex vivo assays, that the canine risk-associated TLR-5 haplotype was characterised by hyper-responsiveness towards flagellin, a common bacterial antigen. Alterations in mucosal mRNA expression, as well as candidate gene mutations, have also been noted for NOD2 in dogs with CIE [29,30].

With respect to acquired immunity, a great deal of work has been done to better define whether CIE is associated with specific cytokine and chemokine patterns. The academic literature on human IBDs has shown differential Th1 (i.e., cell-mediated) and Th2 (i.e., humoral) polarisation in CD and UC patients, respectively [31]. German et al. [32] investigated cytokine expression in mucosal samples obtained from GSDs diagnosed with either idiopathic IBD or ARE and in healthy dogs via semiquantitative reverse transcriptase polymerase. Mucosal mRNA expression of IL-2, IL-5, IL-12p40, TNF-α and TGF-β1 was significantly increased in dogs with CIE relative to healthy controls. Interestingly, antibiotic treatment resulted in reduced TNF-α and TGF-β1 mRNA expression in a subgroup of GSDs affected by ARE. In sharp contrast, a similar study carried out by Peters et al. [33] did not detect any significant difference in IL-2, IL-4, IL-5, IL-6, IL-10, IL-12, IL-18, IFN-γ, TNF-α, TGF-β1 and glyceraldehyde 3-phosphate dehydrogenase duodenal mucosal mRNA expression between dogs with or without CIE. Two distinct reviews on this subject concluded that CIE is generally characterised by an increased, albeit aspecific, cytokine and chemokine expression, thus failing to demonstrate a clear Th1- or Th2-immunological skewing [34,35].

### 2.2. Intestinal Epithelial Barrier

The intestinal epithelial barrier (IEB) is a physical entity made up of a single layer of mucosal cells joined together by multiprotein junctional complexes commonly referred to as tight junctions [36]. The IEB, by governing the passage of luminal antigens across the intestinal epithelium, is crucial for the development of intestinal immunity together with the establishment and maintenance of immunological tolerance [37].

Compelling evidence has been produced regarding the abrogation of mucosal barrier function in CIE. In one study, Sørensen et al. [38] investigated intestinal permeability to selective sugars in 20 dogs affected with either ARE (*n* = 8) or IRE (*n* = 12) and compared it with 20 healthy control dogs. Sugar quantitation was performed on haematological samples by means of tandem high-performance liquid chromatography–pulsed amperometric detection. Lactulose/rhamnose and xylose/3-O-methylglucose ratios in diseased dogs were found to be significantly different from those in the control group, demonstrating the existence of an impaired mucosal permeability. In a similar, controlled study, intestinal permeation to lactulose and rhamnose was evaluated in dogs with lymphocytic–plasmacytic enteritis by measuring the ratio of their urinary concentrations following oral administration of their admixture [39]. The investigators reported a weak correlation between the histopathological grading score of duodenal biopsies and the urinary lactulose/rhamnose ratio; however, subanalysis revealed a far stronger association in those patients showing hypoalbuminemia (<2.5 g/dL) relative to those with normal serum albumin levels. In stark contrast, in a study by Allenspach et al. [40], select urinary sugar ratios (lactulose/rhamnose, xylose/methylglucose, sucrose/methylglucose) did not correlate either to the clinical disease activity or to the histological scoring of intestinal biopsy specimens in CIE dogs.

Immunopathological studies in humans have proved that abrogation of the IEB can progress to bystander damage caused by activation of the aberrant immune response [41]. However, it is legitimate to speculate that any pathological condition leading to augmented intestinal permeability could prime the development of CIE, as suggested by some authors [42].

### 2.3. Intestinal Microbiota and Main Postbiotics

The term “gut microbiota” refers to the consortium of all living microorganisms inhabiting the GI tract of mammalian species. Despite bacteria making up the vast majority of the intestinal microbial biomass, other components of this microscopic ecosystem are represented by fungi, viruses, archaea and protozoa [43].

In dogs, it is well established that bacterial counts augment moving aborally along the GI tract, being the lowest in the stomach (10^1^–10^6^ colony forming units (CFU)/g) and highest in the colon (10^8^–10^11^ CFU/g) [44,45]. Phylogenetic complexity of resident microbial populations follows an overlapping fashion: the large bowel, with its peculiar physicochemical conditions (e.g., plenitude of nutrients, low redox potential), is the segment of the digestive system characterised by the highest microbial richness, being home for hundreds of different phylotypes [46]. In addition to changes along the longitudinal axis of the digestive system, microbial communities may also vary in spatial distribution among different physical niches (e.g., gut lumen, mucus layers, colonic crypts), as shown in humans [47].

Millennia of coevolution have shaped the complex interactions between the host and the gut microbiota, with the latter being involved in a wide array of physiological processes that contribute greatly to the host’s fitness [48]. Indisputably, one of the most important functions attributable to the gut microbiota is the priming and education of the intestinal immune system. Indeed, pioneering studies using germ-free (GF) mice have revealed the anatomical and immunological abnormalities associated with a GF state. In comparison with wild-type animals, GF mice present an underdeveloped immune system, especially at the intestinal level, where fewer plasma cells, smaller Peyer’s patches, a reduced number of mesenteric lymph nodes and impaired Ig-A secretion have been noted [49,50]. Strengthening this causation, it is worth noting how some of the above-cited deficiencies may be reversed following recolonisation with the gut microbiota of a healthy mouse [51].

As shown by extensive research, the finely tuned host–microbiota crosstalk, which modulates systemic and intestinal immunity, is also pivoted on the exchange of microbial products collectively referred to as postbiotics. While some of these metabolites are synthesised de novo by gut microbes (e.g., peptidoglycans and lipopolysaccharides) [52], others represent either intermediates or end-products of the microbial fermentative metabolism. Short-chain fatty acids (SCFAs), chiefly acetate, propionate and butyrate, together with indole, indole derivatives and secondary bile acids, are examples of the latter group [53,54].

SCFAs are saturated organic acids that consist of one to six carbon atoms. Within the large intestine, the vast majority (95–99%) of SCFAs produced are absorbed by the colonic mucosa and stimulate mineral and water uptake from the intestinal lumen [55,56]. One of the primary roles of SCFAs is to serve as energy substrates to fuel host metabolism. Colonic epithelial cells can cover up to 70% of their energy needs via SCFA utilisation, with butyrate being oxidised at a higher rate than propionate and acetate [57]. Intriguingly, it has been conjectured that differentiated colonocytes might avidly metabolise butyric acid in order to protect the stem cells residing in the intestinal crypts from the growth-inhibiting effects of this bacterial metabolite [58]. Moreover, because SCFAs are important in sustaining colonic mucosal metabolism, their shortage in the intestinal lumen could result in mucosal atrophy and, ultimately, in colitis [59]. SCFAs escaping the intestinal mucosal metabolism are conveyed to the liver via the portal vein before reaching the systemic circulation [60]. Comprehensively, it has been estimated that up to 7% of an adult dog maintenance energy requirement can be derived from the oxidation of intestinal microbiota-derived SCFAs [61,62]. Of note, SCFAs can also act as signalling molecules via their bond to cognate nutrient-sensing G-protein coupled receptors [63]. These molecular sensors, which are expressed throughout the GI tract, influence hormone and gut peptide release (e.g., PYY, GLP-1 and GIP), reinforce the gut barrier function, possess immunomodulatory functions and may have a physiological role in regulating intestinal mucosal blood flow and GI motility [64,65,66]. Furthermore, SCFAs have also been demonstrated to hamper the growth of pathogenic bacteria by lowering luminal and faecal pH [67]. Concerning intestinal disorders, the anti-inflammatory capacity of SCFAs has been already documented in vitro, and in vivo in both murine models and human IBDs [68].

At the intestinal level, microbial degradation of luminal nitrogenous substances generates a vast array of metabolites, collectively termed protein fermentation end-products [54]. Unlike carbohydrate-derived postbiotics, these putrefaction compounds are generally associated with detrimental effects on faecal quality and gut health. Indeed, protein fermentation catabolites not only are responsible for faecal odour [69], but also may contribute to the exacerbation of UC in humans [70] and have been associated with an increased risk of colon cancer in rats [71]. Indole, an aromatic heterocyclic organic compound produced from L-tryptophan by certain bacterial species endowed with the enzyme tryptophanase, is one of the major putrefactive compounds [72]. Recent evidence suggests that indole and some of its metabolites (e.g., indole-3-propionic acid and indole-3-acetic acid) can mediate bacterial cell–cell communication but can also work as an interkingdom signal for eukaryotic cells. With respect to the latter, an in vitro study has reported several beneficial effects following the exposure of intestinal epithelial cell colonies to indole [73], such as reduced proinflammatory (IL-8 and TNF-α) and increased anti-inflammatory (IL-10) cytokine production and increased transepithelial resistance together with upregulated expression of genes associated with tight junctions and actin cytoskeleton formation. Conversely, investigations focusing on indole’s functions within the bacterial kingdom have led to controversial results. While on the one hand indole treatment has been shown to hamper microbial biofilm formation [74] and motility [75], on the other it can favour pathogenic bacteria virulence via a set of different mechanisms [76,77,78,79]. Moreover, microbe-derived indoles are passed into the bloodstream and further metabolised by the liver to 3-indoxylsulfate, which represents a known risk factor for cardiovascular and renal diseases in humans [80,81].

Secondary bile acids (SBAs), e.g., deoxycholic acid, lithocholic acid and ursodeoxycholic acid, form another class of postbiotics functioning as signalling modules. SBAs are derived from primary bile acids escaping ileal reabsorption (approximately 5% of total intestinal bile acid pool) through microbial bio-transforming reactions in the large intestine [82]. SBAs are capable of impacting intestinal mucosal immunity via the activation of their receptors, namely the farnesoid X receptor, pregnane X receptor and Takeda G-protein receptor 5 (TGR-5) [83,84]. In a murine study, TGR-5 activation by related bile acid ligands suppressed nuclear factor kappa-light-chain-enhancer of activated B cells (NF-κB)-mediated inflammatory response following lipopolysaccharide injection in wild-type mice but not in TGR-5 knockout mice [85]. In addition, it has been demonstrated that SBA-dependent stimulation of TGR-5 is able to reduce colonic inflammation, arguably by mediating TLR activation pathways [86]. By contrast, SBAs may have detrimental effects (e.g., increased oxidative stress, cellular apoptosis, genotoxicity and mutagenicity) when present in high concentrations [87].

These data underscore the importance of a balanced gut microbiota and optimal metabolism for both intestinal and systemic health. Thereby, microbiota modulation strategies may provide an attractive option to either treat or ameliorate disease severity in CIE.

## 3. Gut Microbiota Alterations in Dogs with CIE

Modifications in gut microbial composition have been hypothesised for a long time to be involved in the pathogenesis of human IBDs and CIE [22,88]. By overcoming the technical limitations of traditional culture-dependent methods, the advent of “-omics” technologies, together with tailor-made bioinformatics tools, have allowed scientists to demonstrate that CIE is associated with intestinal dysbiosis, broadly defined as an imbalance in the composition and functional capacity of the gut microbiota [89,90]. However, whether gut dysbiosis represents a cause or a consequence of mucosal inflammation in dogs with CIE remains a conundrum [91,92].

In a 16S rRNA gene sequencing-based study evaluating the mucosa-adherent duodenal microbiota of IRE dogs, investigators found a lower bacterial richness along with a significantly higher abundance of the *Enterobacteriaceae* family in comparison to healthy dogs [93]. In a similar controlled study, the duodenal mucosa-associated microbiota of dogs suffering from IRE showed an increased abundance of bacteria belonging to Alpha-, Beta- and Gammaproteobacteria classes, whereas members of the Clostridia class were less represented [94]. The mucosa-associated microbiota of dogs affected by IRE was also assessed by Suchodolski et al. [95] via 454-pyrosequencing of the 16S rRNA. In this study, healthy controls showed a higher abundance of the phylum Fusobacteria, order Clostridiales and families *Bacteroidaceae* and *Prevotellaceae*, but lower levels of bacterial genera belonging to the phylum Proteobacteria, relative to dogs with IBD. Moreover, Cassmann et al. [96] investigated the ileal and colonic mucosal microbiota of dogs with different chronic GI diseases via fluorescence in situ hybridisation techniques and found an increased number of *Enterobacteriaceae* (e.g., *Escherichia coli*) either adhering to the epithelial surface or invading the intestinal mucosa in the IBD group when compared to healthy dogs. Similar findings have been reported in human IBD patients, where highly virulent and proinflammatory intracellular *E. coli* strains have been isolated from intestinal tissue [97,98].

While some investigations have evaluated the molecular fingerprint of mucosa-adherent microbial communities, many others have focused on the faecal microbiota composition in dogs with CIE. In this regard, the use of stool is practical due to its ease of collection and the possibility of repeated sampling; however, it is still debated whether the faecal microbiota represents a reliable proxy for gut microbiota and thus for investigation of intestinal dysbiosis in the canine GI tract [99,100]. In a study by Omori et al. [101], stool samples from dogs with either intestinal lymphoma or IBD were analysed by 16S rRNA gene next-generation sequencing and compared to those obtained from healthy individuals. Although rarefaction analysis did not show differences in observed species among the three groups, variations in bacterial composition were detected using principal coordinate analysis in lymphomatous and IBD patients. In more detail, the IBD group showed an increased abundance of the *Paraprevotellaceae* family and the *Porphyromonas* genus compared to healthy dogs. This is of note, as several lines of evidence suggest that intestinal dysbiosis may persist in IBD dogs even after therapy institution and amelioration of clinical signs, with only a partial recovery of the intestinal microbial ecosystem being observed in short-term follow-up [102]. Conversely, treatment response in canines diagnosed with FRE is generally associated with more pronounced effects on the gut microbiota, in terms of both keystone taxa abundance and microbial richness [103,104].

Metabolomics-based research has also been conducted in the field of CIE, shifts in metabolic capacity of the gut microbiota being predictive of intestinal dysbiosis. In a controlled study, Xu et al. [105] evaluated the abundance of a selection of bacterial groups by qPCR and SCFAs and ammonia in stool samples from IBD patients. Despite no significant differences being found, either in terms of bacterial abundance or in terms of fermentative end-products between the two groups, faecal *Lactobacillus* spp. counts and total SCFAs showed an independent negative correlation with the clinical score of IBD. In a separate study, Blake et al. [106] observed a significant decrease in faecal SBAs and a significant increment in faecal lactate concentrations in dogs with CIE relative to healthy controls.

Overall, evidence from cited literature suggests that decreased microbial richness and diversity are two hallmarks of CIE. However, no dysbiosis signature has been identified at present for any of the clinical phenotypes of CIE, since reported compositional alterations in gut microbiota (e.g., higher abundance of *Enterobacteriaceae* and lower abundance of Clostridiales) are common to other canine GI diseases [99,107].

## 4. Main Nonpharmacological Therapies for CIE

### 4.1. Diet

#### 4.1.1. Antigenicity, Digestibility and Nutrient-Responsiveness

Among the known nonpharmacological intervention approaches for CIE, dietetic intervention is acknowledged to be the main conservative treatment [108] and, for this reason, deserves mention in the present review. Epidemiological data from retrospective and prospective clinical studies clearly show that almost two-thirds of dogs diagnosed with idiopathic chronic enteropathies are classified as having FRE, for which full clinical remission can be achieved by dietary therapy alone [109,110,111]. Moreover, dietary manipulation is also advocated, adjuvant to medical treatment, in most other forms of CIE [21].

Reduction of antigenic exposure to GI lymphoid tissue is supposed to play a chief role in mediating the observed diet-associated clinical benefits, given that intestinal inflammation is likely to augment mucosal permeability to luminal antigens (e.g., food-sourced) and prompts oral tolerance breakdown [112,113]. Immunologically speaking, proteins make up the vast majority of food antigens; therefore, great emphasis should be placed on protein-containing dietary ingredients when targeting reduced diet antigenicity. In this regard, the administration of a novel antigen food represents a sound nutritional strategy to avoid the elicitation of acquired immunological hypersensitivity responses [114]. Notwithstanding, sensitisation to the new diet could easily occur in immune-dysregulated patients when intact proteins are fed [115].

In contrast to novel antigen foods, hydrolysed diets typically contain a single protein source that has been cleaved into small polypeptides via enzymatic hydrolysis [116]. As demonstrated by experimental models of type-1 (Ig-E-mediated) food hypersensitivities [117,118], protein hydrolysates are generally characterised by reduced allergenic and antigenic potential over parent compounds, theoretically being more suitable for both short- and long-term nutritional management of CIE. With that being said, extensively hydrolysed proteins may still retain the capability to activate other immunological mechanisms, such as type-4 (lymphocyte-mediated) hypersensitivity, that are thought to be involved in the pathogenesis of a subset of AFRs [119].

Enhanced diet digestibility is another chief goal of a dietetic approach to CIE, as affected patients are often confronted with impaired digestive capabilities. Nutritionwise, the provision of highly bioavailable nutrients allows for counteraction against maldigestion/malabsorption and might ultimately prevent the development of energy-nutrient malnutrition [120]. Furthermore, owing to the lower amounts of major macronutrients escaping assimilation, easily digested foods diminish the intestinal antigenic load and substrates for excessive microbial fermentation in the distal gut. These latter two effects might be accountable for the reported clinical efficacy of highly digestible diets in the treatment of FRE [121]. Despite a lack of a consensus definition at present, easily digestible foods are commonly referred to as having a total apparent tract digestibility of >80% when ingested by healthy subjects [116].

Aside from proteins, dietetic manipulation of fibre or fat has been shown to benefit some dogs with CIE. Despite reducing diet digestibility, fibre fortification may represent a valuable aid in idiopathic chronic colitis owing to the toxin-binding, motility-regulating, water-holding and prebiotic properties of different fibre sources [122]. As far as dietary lipids are concerned, their restriction is generally carried out in patients with (secondary) lymphangiectasia, in order to reduce engorgement of lacteals with chyle, as well as ameliorating steatorrhoea, which can exacerbate diarrhoeal signs via osmotic and secretory mechanisms [123]. Besides, cases of chyle leak may profit from oral supplementation with medium-chain triglycerides, for their being (at least in part) directly absorbed into the portal blood, along with having rapid and simple digestion dynamics [124].

#### 4.1.2. Impact of Diet on Gut Microbiota Composition

Speaking of the mutualistic relationship that exists between the intestinal microbiota and the host organism, the former is fully reliant on the latter for its nourishment. In the proximal intestine, survival of the resident microbiota is based on the digestion and absorption of nutrients [125]. In sharp contrast, the large bowel acts as an anaerobic digester, where inhabiting microbial species obtain energy and carbon skeletons from the fermentation of organic macromolecules, most of which are represented by undigested dietary protein and carbohydrates [126]. Consequently, diet heavily influences the stability and dynamics of the gut microbiota.

Significant alterations in faecal community structure have been documented in canine species following the consumption of diets with a markedly different macronutrient and ingredient composition [127]. For instance, two independent studies reported reduced faecal proportions of the genus *Fusobacterium* (phylum Fusobacteria) and increased proportions of the genus *Faecalibacterium* (phylum Firmicutes) upon dietary fibre supplementation [128,129]. Vice versa, concurrent protein fortification and carbohydrate restriction of diet is generally associated with an overall increase in abundance of the genera *Fusobacterium* and *Clostridium*, whereas members of the *Bacteroides*, *Faecalibacterium* and *Prevotella* genera are reduced [130,131,132]. Modulation of dietary fat content, however, does not seem to affect bacterial diversity of the faecal microbiota, except for a reduced relative abundance of the *Prevotella* genus [133].

The recognition of distinct microbial patterns in dependence on a diet’s macronutrient profile denotes plasticity of the intestinal microbiota, which adapts its metabolic capacity based on available dietary substrates. In humans, population-based studies have found an association between an increased risk of IBD development and the consumption of a high-protein, high-fat, low-fibre diet (i.e., a Western diet). A Western-type diet is likely to promote intestinal inflammation via changes in gut microbiota composition (e.g., increase in Proteobacteria), postbiotic production (reduced SCFAs, augmented protein fermentation metabolites) and host barrier function [134]. Interestingly, similar observations have also been described in dogs. For example, the reduction in faecal *Faecalibacterium* spp. following protein supplementation of the diet might indicate the detrimental effects of dietary protein excess on canine gut health [135], these being members of the *Faecalibacterium* genus provided with anti-inflammatory activity [136]. Besides, dogs receiving a protein-rich diet (i.e., a meat-based diet) have shown reduced faecal SCFA concentrations relative to those fed an extruded food lower in proteins and higher in digestible carbohydrates [130]. Whether these alterations would have any clinical relevance for dogs with CIE, however, remains to be elucidated, and evidence from human medicine could not apply to dogs because of species differences [137].

### 4.2. Phytogenic Feed Additives

#### 4.2.1. Prebiotics

A dietary prebiotic is being referred to as “a substrate that is selectively utilized by host microorganisms conferring a health benefit” [138]. While some studies have given insights on the potential prebiotic properties of dietary compounds not strictly categorised as saccharides [139,140], a large proportion of tested prebiotics in dogs are indeed carbohydrate-based. Prebiotic compounds are found largely in the plant kingdom, where they have structural and energy-storing functions, but can also be isolated from single-celled microorganisms (e.g., yeasts) or from products of animal origin (e.g., milk) or be synthesised enzymatically [141]. From a chemical standpoint, candidate prebiotic compounds are glycans with a variable degree of polymerisation: these comprise monosaccharides (e.g., tagatose), disaccharides (e.g., lactulose, lactitol), oligosaccharides (e.g., short- and long-chain fructooligosaccharides (FOSs), galactooligosaccharides (GOSs), mannan-oligosaccharides (MOSs), xylooligosaccharides, soybean-oligosaccharides, isomaltooligosaccharides) and polysaccharides (e.g., inulin, pectins, resistant starch (RS)) [142,143]. Nondigestibility, which is one of the prerequisites of a substance with prebiotic properties, is attributable chiefly to the presence of β-glycosidic linkages between the sugar residues, whose hydrolysis requires enzymes not produced by the mammalian digestive system but available in the enzymatic repertoire of saccharolytic bacterial symbionts [144].

According to the official definition, the health-promoting effects of prebiotics are thought to be primarily indirect, as being mediated by microbial modulation: upon selective fermentation, prebiotic compounds stimulate the expansion of beneficial, indigenous intestinal bacteria (e.g., lactobacilli, bifidobacteria) and promote their metabolic activity, resulting in the production of several postbiotics, first and foremost SCFAs [138]. Yet, mounting evidence suggests that prebiotic fibres may also affect the host via microbiota-independent modes of action. For instance, certain types of oligosaccharides (e.g., FOSs, GOSs and MOSs) have been shown to block the adhesion of enteropathogenic bacteria to both human and chicken intestinal epithelial cell lines [145,146], likely by functioning as structural mimics of the pathogen binding sites within the GI tract [147]. Moreover, in vitro studies have pointed out that prebiotic fibre sources are able to interact with multiple gut cell types, possibly influencing intestinal immunity and barrier-function in a chain-length-dependent manner [148,149]. Potential molecular mechanisms underlying these effects are represented by the activation of C-type lectin receptors, TLRs and PPAR-γ [150]. Notably, owing to the fact that (small fractions of low molecular weight) prebiotics have been reported to pass the gut barrier intact and enter the systemic circulation [151,152], their direct immunomodulatory actions may extend beyond the digestive system.

The effects of prebiotic administration in healthy dogs have been assessed through a meta-analysis of data reported across 15 studies by Patra [153]. Results showed that total faecal SCFAs, as well as numbers of bifidobacteria and lactobacilli, were positively correlated with increasing supplementation of prebiotic agents. Interestingly, the magnitude of expansion of the above-cited bacteria was comparatively higher when initial bacterial counts were lower. However, prebiotics failed either to significantly reduce the presence of undesired microorganisms (e.g., *C. perfringens*, *E. coli*) or increase serum immunoglobulin (Ig) G, A and M concentrations when administered to healthy subjects. Although nutrient intake, dry matter (DM) and crude fat digestibility were not impaired by increasing dosages of prebiotic compounds, irrespective of the source used, total tract apparent crude protein digestibility tended to decrease in a quadratic fashion. In this regard, it is known that prebiotic-driven increased bacterial protein synthesis in the intestine may be accountable for this finding [154]. The author concluded that feeding prebiotic substances at doses as high as 1.4% (DM basis) seems to be a valid means to beneficially manipulate intestinal microbiota composition and its functionality. In a more recent report, Pinna and Biagi [155] exhaustively reviewed the scientific literature regarding the use of prebiotics in the canine species. In spite of the fact that several inconsistencies were found when comparing study results among each other, the investigators confirmed the positive impact of feeding prebiotics on canine fitness, as testified by an overall enhanced composition of the gut microbial ecosystem, augmented synthesis of SCFAs and mitigated production of certain protein fermentation metabolites. In addition, the same authors inferred that FOSs might be the category of prebiotic compounds performing best in terms of intestinal microbiota manipulation and colonic mineral absorption, whereas MOSs might be ideal candidates for immune function stimulation.

Data proving the effectiveness of prebiotics in chronic intestinal inflammatory processes have emerged from studies involving experimental murine models of colitis [156]. In a set of reports using cohorts of dextran sulfate sodium (DSS)-induced colitic mice, the feeding of different types of prebiotic agents (i.e., inulin, RS, goat milk oligosaccharides, lactulose and FOSs) was able to improve disease activity index (DAI) and reduce colonic inflammation and tissue damage when compared to control groups. With a few exceptions, positive results were also obtained upon oral administration of prebiotics to other experimental models of IBD, such as trinitrobenzene sulfonic acid induced colitis, HLA-B27 transgenic rats and IL-10 knockout mice [157].

In contrast, a paucity of clinical studies testing prebiotics in both human IBDs and CIE have been published at present. In a prospective, randomised, placebo-controlled pilot trial involving human patients with active UC, the administration of oligofructose-enriched inulin in addition to mesalazine was associated with early reduction in faecal calprotectin, a marker of intestinal inflammation [158]. The effects of FOS supplementation on disease activity, faecal and mucosal bifidobacterial concentrations and mucosal dendritic cell function in a cohort of 10 subjects with moderately active CD were investigated by Lindsay et al. [159]. After three weeks of daily FOS administration, the authors reported a significant amelioration of clinical conditions, increased numbers of bifidobacteria in stool and enhanced TLR expression and production of the anti-inflammatory cytokine IL-10 by lamina propria dendritic cells.

In the canine species, Jia et al. [160] evaluated the impact of dietary inclusion of a prebiotic-rich fibre blend (0.2% rice bran, 0.3% banana flakes and 0.4% deactivated yeasts on an as-fed basis) on faecal microbiota composition in a group of nine CIE cases and compared it with healthy controls. Although no differences were observed in terms of stool quality throughout the length of the trial in either of the two arms, 3-week fibre bundle supplementation produced a significant reduction in sulfate-reducing bacteria (i.e., Desulfovibrionales order) whereas numbers of *Clostridium* clusters I and II increased in the CIE group, suggesting a potentially favourable effect on the intestinal microbial ecosystem. A separate, double-blinded, randomised, placebo-controlled trial explored the effects of long-term administration (180 days) of an oral supplement based on chondroitin sulfate and prebiotics (RS, β-glucans and MOSs), along with a hydrolysed diet, in IBD dogs [161]. Outcome measures analysed included clinical signs, intestinal histopathology, faecal microbiota composition and serum biomarkers of inflammation and oxidative stress. Statistical analysis of data from dogs completing the trial (supplement: *n* = 9 dogs; placebo: *n* = 10 dogs) showed no differences in canine IBD activity index (CIBDAI), histological score or faecal microbiota between groups at any time point. However, serum cholesterol and paraoxonase-1 concentrations were found to be significantly higher after 60 days of treatment only in the supplement group, whereas the control arm showed significantly reduced serum total antioxidant capacity levels after 120 days. In a more recent uncontrolled study involving nine food-unresponsive CIE dogs [162], a 30-day administration of a hydrolysed protein food enriched with 4.0% powdered *Ascophyllum nodosum* (a brown seaweed rich in fermentable fibres) was associated with greater faecal amounts of acetic acid and increased numbers of purportedly beneficial bacteria (i.e., *Ruminococcaceae* and *Rikenellaceae* families). Despite that, the dietary treatment failed to ameliorate the patients’ clinical status. Overall, although prebiotics have been shown to modulate the GI microbiota and possibly improve oxidative status, the usefulness of their supplementation in dogs with CIE is yet to be determined.

#### 4.2.2. Phyto- and Phycochemicals

Apart from prebiotics, plants and seaweed contain a bewildering number of naturally occurring, non-nutritive and biologically active compounds, commonly referred to as phytochemicals (plant-derived) or phycochemicals (alga-derived). These substances, which differ greatly in chemical composition, possess different functional properties, such as antioxidant, anti-inflammatory, antimicrobial and immunomodulatory activity [163,164]. Although a sizeable number of both in vitro experiments and preclinical studies with mice have disclosed the therapeutic potential of many different phytochemicals/phycochemicals towards inflammatory intestinal disorders, only the most important molecules (according to the authors’ opinion) will be presented in this section for the sake of brevity.

Diferuloylmethane, also known as curcumin, is a phenolic compound isolated from the rhizome of *Curcuma longa* [165]. Curcumin has become increasingly popular in the research community for its antiphlogistic effects, which are attributable to the phenolic groups found within the molecule [166]. More specifically, it is a potent inhibitor of NF-κB activation, thus blocking the production of proinflammatory cytokines such as IL-1β, -2 and -12 and TNF-α [167,168]. Furthermore, curcumin is also endowed with antimicrobial actions against *E. coli* and *Salmonella typhimurium* [169]. Several studies in different murine models of IBD have found that the administration of curcumin, either orally or systemically, is associated with an improved survival rate and disease clinical score [170]. Positive findings have also been reported in paediatric IBD patients, where a combination of curcumin with standard therapy was associated with a more favourable clinical outcome [171].

Fucoidans are a group of sulfated polysaccharides, composed primarily of L-fucose, which abounds in various brown-seaweed species [172]. The bioactivity type and potency of fucoidans are largely related to their molecular weight, sulfate content and sugar composition [173]. A common feature of many fucoidans is their capability to prevent extravasation of inflammatory cells via P- and L-selectin blockade [174]. In a study by Zhang et al. [175], pretreatment with fucoidans in DSS-induced colitic mice reduced leukocyte extravascular recruitment, thereby attenuating mucosal damage and crypt destruction. Fucoidans may also improve intestinal inflammation via the strengthening of epithelial barrier function, as they have been shown to upregulate the expression of the tight-junction protein claudin-1 [176]. In an ex vivo trial performed in CIE dogs [177], the exposition of intestinal tissue explants to a fucoidan extract from the algal species *A. nodosum* was associated with lower mRNA levels of the proinflammatory genes TNF-α and IL-15, potentially indicating a direct antiphlogistic effect of the tested compound.

Palmitoylethanolamide (PEA) is a fatty-acid ethanolamide that occurs in various foods and is produced endogenously from palmitic acid in mammals [178]. Natural plant sources of PEA are represented mostly by legume seeds (e.g., soybean, peas), although PEA can also be found in certain types of vegetables [179]. From a biological standpoint, PEA functions as a prohomeostatic mediator against inflammation and tissue damage by repressing the activity of both immune cells (e.g., mast cells, monocytes, macrophages) and neuroimmune cells such as astrocytes and microglia [180]. Molecular targets underpinning PEA actions are likely to be diverse and comprise either direct or indirect stimulation of cannabinoid (CB)-1 and CB-2 and PPAR-α receptors [181]. Experimental studies with mice have demonstrated that PEA administration is able to reduce intestinal inflammation and normalise intestinal motility [181,182].

Taken together, scientific evidence on the medical usefulness of phytochemicals and phycochemicals in CIE is limited. Nonetheless, positive data arising from benchtop and clinical research done in other animal species cannot be neglected and make these substances worthy of investigation in the dog as well.

### 4.3. Probiotics

Probiotics are defined as “live microorganisms that, when administered in adequate amounts, confer a health benefit on the host” [183]. In order for a microbe to be termed probiotic, it must satisfy rigorous prerequisites, such as the ability to retain viability during processing, storage and passage through the GI tract; exhibit neither toxicity nor pathogenicity upon administration; and display effects commonly associated with favourable health outcomes. To date, most studied microorganisms with known probiotic properties in the dog are Gram-positive bacteria (e.g., several strains of bifidobacteria, lactobacilli, enterococci and bacilli); however, other nonbacterial organisms, such as the yeast *Saccharomyces boulardii*, could work as potential probiotics [184].

The combined administration of probiotic microbes and substrate(s) selectively utilised by the host microorganisms, which confers a health advantage on the recipient, is described as synbiotic [185]. Based on the modern definition, two different synbiotic approaches exists, namely complementary and synergistic [186]. In complementary synbiotics, component parts are not designed to operate cooperatively and, as such, they must fulfil the evidence and dose requirements for both a probiotic and prebiotic. Conversely, a synbiotic mixture is synergistic in nature whereby the chosen substrate supports the growth or activity of the coadministered microorganisms.

Although yet to be fully unravelled, it is tempting to hypothesise that the mechanisms of action of probiotics are multifarious, plausibly reflecting the marked diversity in their microbiological, compositional and pharmacological attributes [187]. The mechanistic effects of probiotic agents can be clustered broadly into two main groups, namely antagonism against undesired microbial species and modulation of the host’s immune function. In regard to the former category, probiotics are thought to trammel GI colonisation of pathogenic microorganisms by inhibiting their bond to elective adhesion receptors located within the mucus or on epithelial cells [188]. Besides, probiotic agents may favour the creation of a microenvironment that is hostile to certain microbial species, via competition for essential nutrients, the production of antimicrobial substances (e.g., bacteriocins, organic acids) and a strengthening of the intestinal barrier function (e.g., variations in mucus production by goblet cells, increased tight-junction protein expression) [189]. Altogether, these probiotic strategies are referred to as “competitive exclusion” [190]. On the other hand, it is now ascertained that probiotics can interact with the host’s immune defences by means of multiple microbial signals, such as cell-wall components, secretory products and nucleotides [187]. As already detailed, intestinal sensing of microbe-associated antigens (including those of probiotic origin) relies on the recognition by PRRs, whose selective triggering elicits a differential immune response. Notably, evidence from in vitro and ex vivo studies speaks to a marked variation in immunomodulating traits among probiotic organisms regardless of their taxonomic affiliation [191,192]. Furthermore, it has been shown that physical contact or contiguity between intestinal mucosal cells and microorganisms may be critical in inducing the potential immunomodulatory effects, thereby entrenching the importance of adhesiveness in the selection of candidate probiotic microbes [193]. In this context, autochthonous microorganisms are deemed functionally superior to allochthonous strains, by dint of their higher fitness to engraft in the canine GI tract. Notwithstanding, gut colonisation by probiotics appears to be temporary in most cases and dependent on sustained administration, even in the case of commensal microbial species [194,195,196].

In the European Union (EU), the current regulatory framework states that probiotic microorganisms must not pose a risk to either human or animal health [197]. Accordingly, selected microbial strains shall neither bear transmissible drug resistance genes nor produce toxins, virulence determinants or antimicrobial substances that are relevant as antibiotics in human and veterinary medicine [198]. While most microbes used in animal feed are apparently safe, caution should be exercised towards certain microbial groups (e.g., enterococci, bacilli) which might not be exempted from possessing or acquiring pathogenicity traits [199,200].

Quality control should also be of utmost concern when dealing with probiotic agents given that a number of factors can affect their potency. For instance, Grześkowiak et al. [201] found that growth media and pretreatment methods were able to significantly alter the in vitro mucus adhesive ability of three established canine probiotics. Correct identification of incorporated strains and congruent numbers of viable cells per dose are two other core elements to be considered, it being well accepted that effects of probiotics are strain- and concentration-specific. Despite deactivated probiotic microbes being shown to retain some of their functional properties [202], evidence from human studies seems to support the notion that beneficial microorganisms display higher efficacy in live form [203]. Of note is the fact that most probiotic-containing products sold in the EU and North America are categorised as food supplements and, as such, are not legally subject to stringent quality control procedures [184]. The aforementioned regulatory void may account for the variations in veterinary probiotic quality reported by some authors [204].

Pertaining to the health benefits of probiotics in the dog, it is noteworthy that the greater part of the evidence is derived from experimental investigations with healthy animals. In those studies, significant effects have been identified on several outcome measures of gut microbiota composition (e.g., reduced number of pathogenic bacteria) and metabolome (e.g., augmented SCFA production), immune response (e.g., strengthened mucosal immunity by stimulating secretory IgA release), GI function (e.g., improved faecal consistency and nutrient digestibility) and metabolic status (e.g., enhanced lipemic and glycaemic control) [200]. Once again, whether these findings would be of real benefit in the prevention or treatment of intestinal and extraintestinal maladies remains a matter of controversy. Likewise, it is conceivable that a more intensive dosage regimen is needed to evoke the same effects in diseased patients compared to healthy subjects [205].

The rationale for probiotic intervention in CIE stems from the capability of selected microbial strains to influence their underlying tripartite pathophysiological circuit, which involves the intestinal microbiota, mucosal barrier and immune function [187]. At present, however, research output concerning probiotic and synbiotic use in dogs with CIE is rather scant; the relevant literature on the subject is summarised in Table 1.

Published laboratory tests (organ culture models) have focused on probiotic-mediated immunomodulation of the intestinal mucosa in CIE and provide inconclusive evidence. In a pioneering study by Sauter et al. [206], the anti-inflammatory properties of three canine-derived strains of *Lactobacillus* spp., used either singularly or in combination, were assessed in an ex vivo culture system of duodenal biopsies from dogs affected by CIE. Notably, only the probiotic cocktail was able significantly to decrease the ratios of proinflammatory cytokine (TNF-α, IFN-γ, IL-12p40) to regulatory cytokine (IL-10) expression, suggesting that additive or synergistic actions may be achieved by using mixtures of probiotic bacteria. In a separate study, Schmitz et al. [207] investigated the effects of the probiotic strain *Enterococcus faecium* NCIMB 10415 (EF) on cultures of duodenal explants and whole blood from FRE cases. Compared to TLR ligands, EF induced only limited changes in anti-inflammatory gene expression from canine duodenal biopsies, whereas proinflammatory output (TNF-α) from WB was stimulated. The same research team evaluated the intestinal expression of inflammasome components (NLRP3, casp-1, IL-1β and IL-18) in CIE dogs compared to controls when treated with probiotic EF ex vivo and found no significant effect on gene expression in the CIE group [208].

Clinical trials of probiotic use for different CIE subtypes have also been conducted. In a prospective study by Sauter et al. [209], 21 dogs with FRE were allocated randomly to receive either a daily dose of a probiotic cocktail (*L. acidophilus* NCC 2628, *L. acidophilus* NCC 2766, *L. johnsonii* NCC 2767) or a placebo on top of a novel antigen diet for four weeks. Resolution of clinical signs was noted in all animals during the trial, irrespective of treatment received. Moreover, probiotic administration failed to confirm the favourable outcome on cytokine mRNA expression reported in an earlier ex vivo investigation [206]. Another randomised, double-blind, placebo-controlled clinical trial on 12 FRE dogs reported no differences regarding clinical efficacy, histology scores or expression of genes involved in intestinal immunity and barrier function upon supplementation of a hydrolysed elimination diet with a synbiotic product (containing EF, FOSs and gum arabic) for 6 weeks [210]. However, these findings should be interpreted with caution, as the trial was underpowered. In a consecutive study, the effects of the EF-containing synbiotic on gut microbiota composition were investigated [211]. Although FRE dogs displayed a small increase in faecal species diversity at the end of synbiotic treatment, there was no significant difference in microbial community composition between groups.

With reference to ARE, the prophylactic and therapeutic role of probiotics was evaluated in a group of nine dogs diagnosed with tylosin-responsive diarrhoea [8]. In this uncontrolled clinical trial, treatment with *L. rhamnosus* ATCC 53103 strain (LGG) was instituted directly after tylosin discontinuation, with patients being followed for up to 30 days. Unlike antibiotic therapy, LGG intervention failed to prolong the asymptomatic period as diarrhoea relapsed after 3–26 (mean 7) days in all enrollees. Another uncontrolled study with food-unresponsive CIE cases (*n* = 6 ARE and *n* = 3 IRE) investigated the efficacy of a high-dose probiotic supplementation, based on *Bacillus subtilis* DSM 15544, for a total of 30 days [162]. Even though the treatment tended to increase faecal concentrations of butyrate, no improvements were recorded in terms of median CIBDAI scores.

Lastly, probiotic-based investigations have been also conducted in dogs with idiopathic IBD. A 90-day randomised, comparative study by Rossi et al. [212] assessed the microbiological, histological and immunomodulatory effects of a high concentration multistrain probiotic preparation (VSL#3, comprising four strains of *Lactobacillus* spp., three strains of *Bifidobacterium* spp. and one strain of *Streptococcus salivarius* subsp. *thermophilus*) over a combination therapy with prednisone and metronidazole in 20 IRE dogs. Despite a more rapid recovery being noted in animals receiving pharmacological treatment, both treatment arms showed an equal improvement in terms of clinical outcome, duodenal histology scores and reduced numbers of proinflammatory CD3+ cells at the end of the trial. Besides, the VSL#3-treated group showed an increase in FoxP3+ immunosuppressive cells in bioptic samples and a higher abundance of the genus *Faecalibacterium* after treatment, indicating a protective effect towards intestinal inflammation. In a more recent, controlled study with 34 IRE patients, the same probiotic cocktail was tested on top of standard therapy (i.e., elimination diet and oral prednisone) for a total of 8 weeks [213]. Both treatments were associated with rapid clinical remission and a higher microbial richness within adherent mucus, although no improvement in histopathologic inflammation was noted. Notwithstanding, an upregulation of tight-junction protein expression was observed only in the probiotic-supplemented dogs, which might denote a beneficial effect of administered bacterial strains on mucosal homeostasis. In a trial by D’Angelo et al. [214], 20 canine patients with IBD were administered the probiotic yeast *S. boulardii* or a placebo in addition to diet (commercial novel antigen, hydrolysed or restricted home-cooked), antibiotics (oral tylosin or metronidazole), steroids (oral prednisone) and/or other immunosuppressants (e.g., oral azathioprine or chlorambucil) and followed for 60 days. Only 13 dogs reached the end of the study, 6 of which were in the probiotic group. While no significant differences were noticed regarding ultrasound, endoscopic or histopathological appearance between the two groups, *S. boulardii* treatment was well tolerated and associated with a significant improvement in clinical activity index, stool frequency, stool consistency and body condition score compared to placebo. Altogether, data from interventional probiotic studies in dogs suffering from CIE are controversial. Moreover, the comparability and generalisability of study results are hindered by differences in experimental setting, study population and microbial strains tested.

### 4.4. Faecal Microbiota Transplantation

The term faecal microbiota transplantation (FMT), also known as faecal bacteriotherapy or stool transplant, refers to the perfusion of distal faecal matter from a healthy donor into the intestinal tract of a recipient in order to confer a health benefit [215]. The first historical report of gut microbiota transplant in animals dates back to the 17th century, when the Italian anatomist Fabricius Aquapendente used it by the name of “transfaunation” to treat rumination disorders in cows [216]. While traditionally being a common practice in livestock, the use of FMT in humans and in domestic carnivores has gained *momentum* only in recent times, after it proved to be a valuable therapeutic modality for the treatment of human recurrent *Clostridioides difficile* infections [217,218].

Advances in gut microbiota profiling have further expanded the clinical frontiers of stool transplant, providing evidence that a multitude of digestive and extradigestive infirmities are associated with a permanent intestinal dysbiotic status [219,220,221]. That being said, it is not surprising how FMT-related scientific output has experienced a boom since the beginning of the 2000s and the trend continues upwards [222].

The exact mechanisms whereby stool transplants affect the recipient’s biology are yet to be fully elucidated. Notwithstanding, it is legitimate to speculate that restoration of a balanced gut microbiota composition and related metabolic function is likely to play a primary role in the clinical success of FMT [223,224]. In support of this notion, long-term engraftment of donor microbiota and functional changes in the gut microbial metabolome have already been demonstrated in humans [225]. For this reason, FMT can be regarded as a probiotic intervention “*in extremo*”. As microbial viability is indispensable in allowing donor microbiota engraftment, standardisation of FMT handling and storage procedures are of the utmost importance in impacting positively on the survival of certain bacterial species (i.e., strict anaerobes) that are likely to be key for therapeutic efficacy [226,227]. An additional therapeutic driver of faecal bacteriotherapy consists of a direct interaction with the recipient’s immune system. Unprocessed stool is indeed a rather complex, biologically active matrix, comprising billions of different microbes, colonocytes and a wide range of microbial metabolites (e.g., SCFAs), most of which have been demonstrated to possess immunomodulatory properties [228].

Studies on murine IBD models have offered the opportunity to investigate, in a tightly controlled experimental setting, the therapeutic potential of FMT. In a study by Tian et al. [229], reduced DAI and reduced colon inflammation, as well as decreased inflammatory cytokine levels, were found in DSS-induced colitic BALB/c mice after intracolonic injection of 150 μL faecal suspension once daily for a total of 8 days. In a similar study, Zhou et al. [230] assessed the efficacy of stool transplant in a mouse surrogate of DSS-induced colitis and compared it with that of 5-aminosalicylic acid (5 ASA) during an 8-day trial. Mice were given either 200 μL faecal slurry by enema or 5 ASA in suspension (100 mg/kg) on days 1, 3, 5 and 7. Although 5 ASA performed slightly better in reducing some inflammatory parameters, both treatments were equally efficacious in the treatment of UC. Notably, there is evidence that the transfer of a “colitogenic microbiota” from UC mice to cohoused healthy individuals is able to elicit the disease [231].

In dogs, FMT has been tested in either the treatment or prevention of a variety of digestive disorders, first and foremost in cases of CIE [232,233,234]. However, it must be borne in mind that the vast majority of the available scientific literature on the subject in question is represented by small-scale, uncontrolled trials that used different FMT protocols. In a case series comprising 16 dogs diagnosed with NRE, Bottero et al. [235] performed FMT along with dietary (novel antigen, highly digestible or hydrolysed protein diet) and pharmacological (antibiotic and/or immunosuppressor) therapy. Stool transplant was performed either endoscopically as a single infusion of 10 mL/kg body weight (BW) faecal suspension and/or orally (1.5–3 g per animal per day). The authors reported a reduced canine chronic enteropathy clinical activity index (CCECAI) at 1-month follow-up, with a better long-term (3-month) clinical outcome when endoscopic FMT was followed by a daily oral dose of frozen donor stool as maintenance therapy. A case report by Niina et al. [236] explored the clinical and microbiological effects of long-term stool transplant in a canine patient affected by NRE. Periodic infusion of 10 mL/kg BW faecal suspension by enema for a total of 180 days resulted in induction and maintenance of clinical remission, as testified by a significant reduction in CIBDAI. Moreover, relative to a baseline sample, the post-treatment faecal microbiome clustered phylogenetically with that of the donor, denoting normalisation of the recipient’s gut microbiota. Remarkably, FMT was well tolerated and stabilised the animal’s stool consistency for up to 63 days following its inoculation. Similar results were reported by Berlanda et al. [237], who carried out orally delivered FMT in a 9-year-old IBD dog. In this study, two 30-day cycles of concentrated lyophilised stool (1 capsule per day), given 8 months apart, were associated with an amelioration of GI symptomatology, along with a gradual shift of microbiome parameters to values similar to those of healthy animals. In a more recent uncontrolled trial, Niina et al. [238] administered stool transplant rectally at a dosage of 10 mL faecal slurry/kg BW, to a cohort of nine dogs diagnosed with IRE. Even though medications were discontinued prior to FMT, the patients showed a significant improvement of disease activity as soon as 3 days following the procedure and maintained clinical response until the 2-week follow-up. Comprehensively, although there seems to be some evidence to indicate the usefulness of stool transplant in CIE treatment, further work is required in order to determine the best FMT methodology and administration protocols, as well as to define its indications and safety aspects.

### 4.5. Stem Cell Therapy

Stem cell therapy is defined as “direct or indirect (derivation) use of different types of stem cells from different sources for therapeutic purposes” [239]. In canine cell-based therapies, mesenchymal stem cells (MSCs) are the most studied cell type due to their wide distribution in animal tissues (e.g., bone marrow, adipose tissue), ease of harvesting and expansion in culture [240]. MSCs are endowed with unique functional attributes, such as the capability to migrate to injury site(s) when administered systemically (i.e., homing) and to replicate and differentiate into diverse cell lineages, eventually leading to tissue regeneration [241]. However, a large and growing body of research points out that the healing properties of MSCs may primarily be related to anti-inflammatory, immunomodulatory and trophic actions, which occur either via cell-to-cell contact or in a paracrine fashion [242]. Interestingly, hundreds of different soluble proteins and vesicular factors, collectively termed “secretome”, are released by MSCs; secreted proteins include bioactive molecules such as immunoregulatory cytokines (e.g., IL-10, TNF-α, TGF-β1) and chemokines (e.g., eotaxin-3), as well as growth factors (e.g., VEGF, hepatocyte growth factor) [243].

Because of their pleiotropic biological effects, MSCs represent an attractive therapeutic avenue for a variety of clinical conditions, including chronic intestinal diseases [244,245]. The effects of canine MSCs have been tested in xeno-transplantation studies involving a murine model of IBD. In a report by Song et al. [246], intraperitoneal administration of 2 × 10^6^ canine adipose-tissue-derived (cAT)-MSCs to DSS-induced colitis resulted in significantly reduced BW loss, DAI and shortening of colon length in comparison to mice treated with phosphate-buffered saline. Furthermore, two other distinct studies demonstrated that priming cAT-MSCs with proinflammatory cytokines (e.g., TNF-α, IFN-γ) prior to infusion boosts the secretion of immunomodulatory factors and thereby is associated with a better clinical outcome in experimental colitis mice compared to nonstimulated cAT-MSCs [247,248].

To the authors’ knowledge, to date, only two clinical trials have looked at the feasibility and safety of stem cell therapy in dogs with CIE. In a study by Pérez-Merino et al. [249,250], 11 dogs with a confirmed diagnosis of IBD received a single intravenous dose of allogeneic, single-donor cAT-MSCs (2 × 10^6^ cells/kg BW) and were followed up weekly for 6 weeks for clinical activity indices and biochemistry parameters and for 90–120 days for endoscopic and histological scores. No acute reaction to cAT-MSC injection and no side effects were recorded throughout the observation period in any patient. By day 42, a significant reduction in both CIBDAI and CCECAI was reported, whereas serum albumin, cobalamin and folate concentrations increased substantially. In addition, significant differences between pre- and post-treatment were also found regarding histological and endoscopic measures. A more recent comparative trial by Cristóbal et al. [251] investigated the long-term effects of allogeneic cAT-MSC infusion at a dose of 4 × 10^6^ cells/kg BW, with or without concurrent prednisone treatment, in a cohort of 32 NRE canine patients. In both groups, clinical scores and serum albumin and cobalamin concentrations improved progressively at each time point, over a total of 12 months. Moreover, the treatment was well tolerated by all the animals and allowed a gradual discontinuation of steroidal therapy in prednisone-cotreated dogs by the end of the observation period. In spite of the encouraging results, a considerable gap of knowledge needs to be filled regarding the best practices of selection, laboratory preparation and administration of stem cells to treat CIE.

## 5. Conclusions

The present review set out to evaluate emerging nonconventional therapies for the management of CIE. The postulated mechanisms of action of the nonpharmacological therapies reviewed are summarised in Figure 1. Although promising, the current literature regarding the efficacy of novel treatment agents in CIE is rather scarce and thus inconclusive. As such, the conduction of properly designed, adequately powered clinical trials is urgently warranted in order to improve the generalisability of study results. With respect to microbial manipulation strategies, future advances in the profiling of GI microbiota, as well as clarification of its exact pathophysiological role in different CIE phenotypes, will ultimately allow for a more targeted intervention, thereby increasing therapeutic yield. Lastly, attention should be paid to investigating potential interactions between drugs and nonpharmacological resources in the case of combination therapy, as already highlighted by evidence from human medicine.

## Figures and Tables

**Figure 1 vetsci-09-00037-f001:**
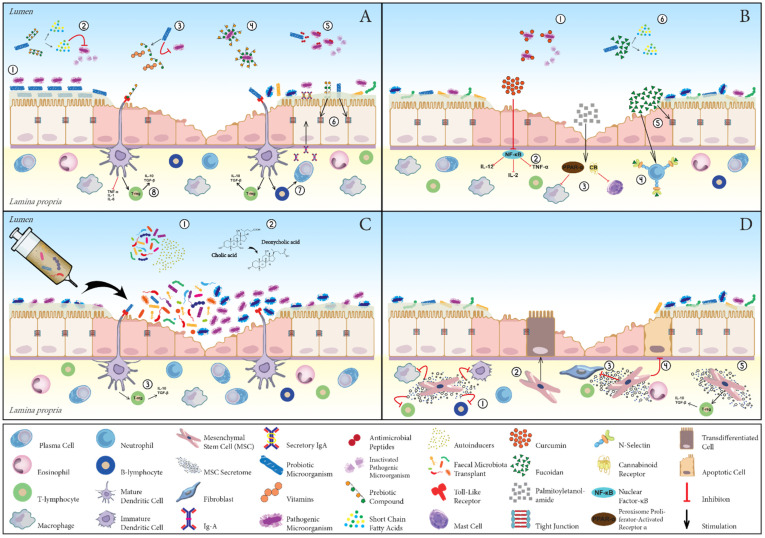
Schematic representation of the potential mechanisms underpinning beneficial effects of the main nonpharmacological treatment strategies in chronic inflammatory enteropathy of dogs (CIE). (**A**) Pre-, pro- and synbiotics: Probiotic microorganisms compete with pathogenic bacteria for adhesion sites at the level of the mucus layer or onto enterocytes (1); ferment prebiotic fibre with consequent production of postbiotics (e.g., short-chain fatty acids) (2); compete for growth substrates and produce and release essential dietary nutrients (e.g., vitamins) (3); inhibit the expansion of pathogenic microorganisms via the production of antimicrobial peptides (5); promote, together with prebiotic fibre compounds, tight-junction protein expression and strengthening of gut barrier function (6); and modulate immune responses by expanding the population of regulatory T cells (T-regs) and enhancing secretory immunoglobulin-A production (7). Additional direct mechanisms of action of prebiotics encompass blockage of pathogen adhesion by serving as ligand analogues (4) and immunoregulatory effect, exerted via the differential activation of inflammation-related receptors (8). (**B**) Phyto- and phycochemicals: curcumin is endowed with antibacterial activity against deleterious microbes (1) and functions as a potent anti-inflammatory compound by inhibiting nuclear factor-κB (NF-κB) (2); palmitoylethanolamide (PEA) quenches phlogistic reactions through the activation of cannabinoid (CB) receptors and peroxisome proliferator-activated receptor (PPAR)-α (3); fucoidan hinders inflammatory cell egression from blood vessels via P- and L-selectin blockade (4), upregulates tight-junction protein claudin-1 expression (5) and is a source of fermentable fibre (6). (**C**) Faecal microbiota transplant: stool transplant reinstates gut homeostasis through a direct interaction between donor and recipient intestinal microbiota (1), restores normal faecal bile acid metabolism (2) and exerts immunoregulatory effects (3). (**D**) Stem cells: mesenchymal stromal cells suppress the activity of different immune cells (1), transdifferentiate into enterocytes (2), hamper fibrosis (3) and intestinal epithelial cell apoptosis (4) and induce T-reg differentiation and expansion (5).

**Table 1 vetsci-09-00037-t001:** Specifications of selected studies evaluating the efficacy of probiotic/synbiotic treatment in dogs with CIE.

Reference	Inclusion Diagnosis	Experimental Setting *	Probiotic Strain(s)/Treatment	Probiotic Dosage	Time ^§^	Main Outcomes
Sauter et al. (2005) [206]	CIE	Ex vivo study	*L. acidophilus* NCC 2628, *L. acidophilus* NCC 2766, *L. johnsonii* NCC 2767	1 × 10^7^ CFU/mL of medium	36 h	Increased IL-10 mRNA and protein expression; decreased ratio of TNF-α/IL-10, IFN-γ/IL-10 and IL-12p40/IL-10 mRNA levels.
Schmitz et al. (2014) [207]	FRE	Ex vivo study	*E. faecium* NCIMB 10415	1 × 10^7^ CFU/mL of medium	5 h	Increased TNF-α protein expression from whole blood in both groups. TNF-α protein responses opposite in blood and biopsies.
Schmitz et al. (2015b) [208]	CIE	Ex vivo study	*E. faecium* NCIMB 10415	1 × 10^7^ CFU/mL of medium	5 h	No effect on NLRP3, casp-1, IL-1β and IL-18 gene and protein expression.
	FRE	In vivo placebo-controlled randomised trial	*E. faecium* NCIMB 10415 + FOSs + gum Arabic + hydrolysed protein diet	1 × 10^9^ CFU/dog/day	42 days	
Sauter et al. (2006) [209]	FRE	In vivo placebo-controlled randomised trial	*L. acidophilus* NCC 2628, *L. acidophilus* NCC 2766, *L. johnsonii* NCC 2767 + novel protein diet	1 × 10^10^ CFU/dog/day (of each strain)	28 days	Decreased duodenal IL-10 and increased colonic IFN-γ mRNA expression; ^†^ increased numbers of *Lactobacillus* spp.; ^†^ detection of *L. johnsonii* NCC 2767 in 5 of 8 dogs after probiotic supplementation; no significant differences in clinical response between groups.
Schmitz et al. (2015a) [210]	FRE	In vivo placebo-controlled randomised trial	*E. faecium* NCIMB 10415 + FOSs + gum Arabic + hydrolysed protein diet	1 × 10^9^ CFU/dog/day	42 days	No significant differences in clinical efficacy and histology score between groups. No effect on TLR-2, -4, -5, -9; IL-17A; IL-22; IL-23p19; RORC; IL-2; IL-12p35; TNF-α; IL-4; IFN-γ; IL-10; TGF β; IL-1β; IL-18; NLRP3; casp-1; TFF1; TFF3 and PPAR-γ mRNA expression.
Pilla et al. (2019) [211]	FRE	In vivo placebo-controlled randomised trial	*E. faecium* NCIMB 10415 + FOSs + gum Arabic + hydrolysed protein diet	1 × 10^9^ CFU/dog/day	42 days	Small increase in faecal species diversity; no significant differences in microbial community composition between groups.
Westermarck et al. (2005) [8]	ARE	In vivo uncontrolled study	*L. rhamnosus* ATCC 53103	1 × 10^10^ CFU/dog/day	≤30 days	Failure to avoid recurrence of diarrhoea in 9 of 9 dogs.
Isidori et al. (2021) [162]	ARE + IRE	In vivo uncontrolled study	*B. subtilis* DSM 15544	125 × 10^9^ CFU/10 kg BW/day	30 days	No significant differences in clinical outcome between pre- and post-treatment. Increased faecal concentrations of butyric acid. ^†^
Rossi et al. (2014) [212]	IRE	In vivo comparative randomised trial	*L. plantarum* DSM 24730, *S. thermophiles* DSM 24731, *B. breve* DSM 24732, *L. paracasei* DSM 24733, *L. delbrueckii* subsp. *bulgaricus* DSM 24734, *L. acidophilus* DSM 24735, *B. longum* DSM 24736, *B. infantis* DSM 24737	112–225 × 10^9^ CFU/10 kg BW/day	60 days	Decreased clinical and histological scores and reduced proinflammatory CD3+ T-cell infiltration in both study groups; increased FoxP3+ immunosuppressive cells and relative abundance of genus *Faecalibacterium*.
White et al. (2017) [213]	IRE	In vivo placebo-controlled randomised trial	*L. plantarum* DSM 24730, *S. thermophiles* DSM 24731, *B. breve* DSM 24732, *L. paracasei* DSM 24733, *L. delbrueckii* subsp. *bulgaricus* DSM 24734, *L. acidophilus* DSM 24735, *B. longum* DSM 24736, *B. infantis* DSM 24737 + prednisone + elimination diet	112–225 × 10^9^ CFU/10 kg BW/day	56 days	Increased E-cadherin, occludin and zonulin protein expression.
D’Angelo et al. (2018) [214]	IRE	In vivo placebo-controlled nonrandomised trial	*S. boulardii* + dietary therapy + antibiotics + steroids ± immunosuppressors	1 × 10^9^ CFU/kg BW/twice a day	60 days	Lower clinical activity index, stool frequency, stool consistency; higher body condition score.

^§^ Incubation time lapse for ex vivo studies; ^†^ tendency (0.05 < *p* ≤ 0.1); ARE = antibiotic-responsive enteropathy; BW = body weight; CIE = chronic inflammatory enteropathy; CFU = colony-forming units; FOSs = fructooligosaccharides; FRE = food-responsive enteropathy; IRE = immunosuppressive-responsive enteropathy. * Ex vivo studies were performed on freshly retrieved duodenal explants from dogs with CIE. Group divisions per reference: [206]: healthy dogs vs. dogs with CIE evaluated before and after exposure to lactobacilli; [207]: healthy dogs vs. dogs with FRE evaluated before and after *E. faecium* exposure; [208]: healthy dogs vs. dogs with CIE, with CIE group receiving hypoallergenic diet either alone or in combination with the symbiotic product.

## Data Availability

The data presented in this study are available on request from the corresponding author.

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
