# Peer review of "Nonpharmacological Treatment Strategies for the Management of Canine Chronic Inflammatory Enteropathy—A Narrative Review"

_vetsci, 2022, doi:10.3390/vetsci9020037_

Round 1

Reviewer 1 Report

Non-pharmacologic ....(Isidori et al.)

This very long and well documented article (235 bibliographic references) represents a very substantial review of the literature on a very important subject in veterinary medicine.

After a short introduction that aims to define the subject, the author reviews the different elements that can explain the disease (ethiopathogenesis, immunity, intestinal barrier and the role of the microbiota and its alterations). He then explores non-pharmacological therapies, namely diet and its impact, phytogenic additives (including pre-biotics), probiotics (with a summary table), and fecal transplantation.

It is not uninteresting but a little long and finally concludes that there is a lack of experimental or clinical research in dogs.  Regarding probiotics, the author should mention the studies that show their lack of efficacy if added to food; the only conclusive studies have been with supplements.

The article is well written; everything is understandable.  It is also fairly well structured, with results from mouse or human models clearly identified in separate paragraphs from discussions of the dog (lines 458, 467, ...) The conclusion is surprising, however, given the title of the article, it contains no recommendations for clinically applied treatment; there is no answer to the underlying question.  It would still be desirable to draft some form of management recommendation.

It is an article of interest to the general veterinarian but provides little information for the gastrology specialist who will be much more informed about current treatments or studies.   The form is certainly narrative but for the practitioner there is an interest in having practical information.

Minor comments

Figure 1, which illustrates the modes of action at the intestinal level, is too crowded and therefore not very readable in its current form. 

Similarly, Table 1 should be presented in "landscape" mode as it is not very readable due to the amount of detail it contains.

Line 175: the term "fitness" is not the most appropriate

Line 478: it would be interesting to point out that the borderline between prebiotics and normal food components is sometimes very thin and this is particularly the case for the example presented

The size (and shape) of the characters is not harmonised throughout (faecal microbial transplantation or stem cell therapy are in italics, for example).

Reviewer 2 Report

The review article ‘Non-pharmacological treatment strategies for the management of canine chronic inflammatory enteropathy – a narrative review ‘ contains highly up to date content and demonstrates a vast knowledge of the literature as well as of the gaps in the field. It addresses a compelling topic and is broad in scope and original at that. The manuscript is very organized, coherent and well written, although it sometimes requires the reader to have some background knowledge on the matter. At times, specific terms are chosen (such as for example ‘foxp3+ cells’ or ‘CD3+ cells’) that could be clarified in order to facilitate the reading. In addition, some parts are repetitive and can be omitted.

I have some minor remarks/suggestions:

-Line 6: Typo in email address massimo.trabalzmarinucci@unipg.it: ‘a’ is missing

-Line 11-29: please also include syn- and postbiotics in the abstract, as discussed in the manuscript

-Line 12: Please consider changing to ‘If conventional dietary treatment alone would be unsuccessful, management of CIE is …’ Please consider using ‘conventional’ dietary treatment in this sentence as to differentiate with the use of pre-, pro-, syn- and postbiotics, FMT, …

-Line 16: ‘Therefore, the development of … is highly sought after.’ -> Tautology, please consider changing to: ‘Therefore, novel, safe and effective therapies for CIE are highly sought after.’

-Line 17-19: Please consider changing this sentence to: ‘As gut microbiota imbalances are often associated with GI disorders, a compelling rationale exists for the use of non-pharmacological methods of microbial manipulation in CIE, such as faecal microbiota transplantation and administration of pre-, pro-, syn- and postbiotics.’ -there are some indications that postbiotics could possess certain feedback mechanisms and might in fact be able to influence microbiome composition and function as do prebiotics (Loh et al., 2013; Klemashevich et al., 2014)-

-Line 26: Methodology is the study of research methods, don’t you mean ‘mode of action’ here?

-Line 34: ‘and they are’ <-> CIE is singular at the start of the sentence

-Line 36-37: ‘antibiotic responsive enteropathy’: abbreviation is missing (used later on in the text)

-Line 40: please add reference

-Line 58-59: please consider rephrasing -> or might develop into intestinal neoplasms due to a sustained impairment of host defences.

-Line 74: is critically to appraise -> please change to ‘is to critically appraise’

-Paragraph 79-83: please add reference

-Line 98: plural, the abbreviation would be PAMPs

-Line 94-99: please add reference

-Line 101-103: please clarify that this was a canine study

-Line 115: to define better -> please change to ‘to better define’

-Line 117: please consider clarifying Th1/17/2 (cfr. explanatory introduction in the innate immunity section)

-Line 131: the intestinal barrier consists of more components than just the IEB, please consider changing the title of this section or including a paragraph regarding mucus and glycocalyx

-Line 134: not all passage of antigens is transcellular at this level

-Line 155: ‘Although’ is probably not the right term here

-Line 184-186: please add reference

-Line 191-194: please add reference

-Line 200-203: please also consider the manuscript by Kaiko et al., 2016: it is possible that colonocytes consume butyrate as a source of energy in order to protect the stem cells residing in the intestinal crypts from this growth-inhibiting metabolite.

-Line 200-20”: ‘primary roles of SCFA’ and ‘the fraction of SCFA’ -> this makes it seem as if the majority of SCFA produced would be butyric acid, which would not be correct.

-Line 213: please also adress the potential use of postbiotics in IBD cases (for example Klampfer et al., 2003)

-Line 230: Protein catabolites are not only responsible for faecal odor, but are also known to worsen intestinal disorders in humans (Ramakrishna et al., 1991) and are associated with an increased risk of developing colon cancer in rats (Lin and Visek, 1991).

-Line 246: not metabolomics, but metabolism

-Line 254: ‘Intestinal dysbiosis, broadly defined as an imbalance in the composition and functional capacity of the gut microbiota, in dogs suffering from chronic enteropathies [80,81]’ -> implies that intestinal dysbiosis requires CIE, please rephrase

-Line 256-259: please remove second half of the sentence, double

-Line 342-361: please integrate into one paragraph (one sentence e.g. 359-361 can never constitute an entire paragraph)

-Line 359: ‘On the other hand’: please rephrase -> implies a discrepancy with the previous sentence

-Line 369: in -> influence

-Line 395: please clarify: ‘extruded food’ is not really a nutrient profile (extruded diets can also be high in protein and lower in carbohydrates -please differentiate between non-structural carbohydrates and fibre here-)

-Line 403: please remove the comma after ‘changes’

-Line 415: please add bracket after isomalto-oligosaccharides

-Line 411-416: please one sentence can never constitute an entire paragraph, the same goes for 402-404. Please consider to combine the paragraphs.

-Line 417-424: Please consider changing the order of these two sentences.

- Paragraph 425-431: please add reference. ‘The alleged health-promoting effects of prebiotics are predominantly indirect and microbial-mediated’ -> is this really correct? (see Speert et al., 1984; Davis et al., 2004; Vogt et al., 2014, Lehmann et al., 2015;…)

-Line 429-430: remove ‘since they play pivotal physiological roles at both intestinal and systemic levels.’  -> double

-Line 445: ‘to manipulate beneficially intestinal microbiota composition’ -> please change to ‘to beneficially manipulate…’

-Line 450-452: ‘as testified by … and decreased production of protein fermentation metabolites’. This is in contrast with the beneficial effects of indole discussed in line 214

-Line 470: Please change ‘on top of’ to ‘in addition to’

-Line 476: Clarifying that lL-10 is anti-inflammatory cytokine would facilitate reading here

-Line 478-481: please specify that faecal samples were analysed in this particular study: gut microbiota composition -> faecal microbiota composition

-Line 503: please also address the direct effect of prebiotics on host immune system (both at local intestinal level and systemically -small fractions of ingested oligosaccharides have been demonstrated to pass the gut barrier, entering the systemic circulation and being recovered in urine of human subjects for example (Molis et al., 1996)-

-Line 596: ‘as well as not being able to’ please rephrase

-Line 604: please change to ‘were able to significantly alter’

-Line 616-622: Please provide references for the different studies (reference 181 does not seem correct).

-Table 1:

-I would consider removing this table, as this is discussed extensively in the text already (or maybe move to supplemental material if possible?). If kept in, please consider the following:

-Sauter et al, 2005: please specify in the table that this concerns duodenal samples

-please clarify the different groups in the ex vivo trials

-Schmitz et al. (2014): responses to bacterial stimuli differ by location: TNFα protein responses in blood were opposite of what was seen in the biopsies.

-Line 633: please define tendency and add a comma here

-Line 663: please make a different chapter on synbiotics, in which you may also speculate on the possible synergistic effects of pre-and probiotics

-Line 692: ‘to receive, or not,’ please rephrase

-Line 734: has-> have

-Fig 1: I would add the direct effect of prebiotics on host immune system, as well as synbiotics and postbiotics in this figure. Secretory immunoglobulin-A: first mention of this compound in the conclusion, please discuss in the main text as well.

Reviewer 3 Report

Dear authors,

I consider your work an excellent contribution to the systematization of knowledge about non-pharmacological alternatives to the treatment of chronic inflammatory bowel diseases.

I particularly liked the summary figure and the use of tables regarding probiotics, which in my opinion should have been extended to other topics.

I just leave a few comments in order to contribute to a definitive version without some typos and that reflects some aspects not considered in this version.

Lines 36 to 37 - antibiotic responsive (ARE)

Lines 44 to 47 - In line 470 you refer to mesalazine which does not belong to any of the groups presented in lines 44 to 47, being an anti-inflammatory.

4.1. Diet 
4.1.1. Antigenicity, digestibility and nutrient-responsiveness

Line 355 - The reference to the use of medium chain triglycerides (MCT) in this context seems to me to be adequate.

Line 388 - post-biotic but Line 160 - postbiotics

Lines 455 to 457 - It would be important to mention this dimension of the MOS.

"The use of MOS to block pathogen colonization derives from the conception that certain polysaccharides could be used to block the mechanism of recognition and adhesion of potential pathogens to molecules on the surfaces of host tissues (competition for attachment sites). This action would reduce the adhesion of the pathogens to the digestive tract, leaving them to be excreted in the feces. This may lead to the improvement of the integrity and performance of the intestinal epithelial barrier"

5. Clinical summary and remarks for the future

clinical summary?

Line 838 - dissertation ?
